# A Highly-Efficient Group Elastic Net Algorithm with an Application to Function-On-Scalar Regression

**Tobia Boschi**
Department of Statistics
Penn State University
tub37@psu.edu

**Matthew Reimherr**
Department of Statistics
Penn State University
mlr36@psu.edu

**Francesca Chiaromonte**
Department of Statistics
Penn State University,
EMbeDS
Sant'Anna School of
Advanced Studies
fxc11@psu.edu

## Abstract

Feature Selection and Functional Data Analysis are two dynamic areas of research, with important applications in the analysis of large and complex data sets. Straddling these two areas, we propose a new highly efficient algorithm to perform Group Elastic Net with application to function-on-scalar feature selection, where a functional response is modeled against a very large number of potential scalar predictors. First, we introduce a new algorithm to solve Group Elastic Net in ultra-high dimensional settings, which exploits the sparsity structure of the Augmented Lagrangian to greatly reduce computational burden. Next, taking advantage of the properties of Functional Principal Components, we extend our algorithm to the function-on-scalar regression framework. We use simulations to demonstrate the CPU time gains afforded by our approach compared to its best existing competitors, and present an application to data from a Genome Wide Association Study on childhood obesity.

## 1   Introduction

As problems involving very large and potentially structured data become ever more ubiquitous, attention is being devoted to the integration of approaches and techniques from the areas of Feature Selection and Functional Data Analysis (FDA). Indeed, more and more regression applications comprise a large number number of variables – some of which are scalar and some of which are suitable for a functional representation, such as longitudinal measurements or biomedical images (Sørensen et al., 2013; Ullah and Finch, 2013; Cremona et al., 2019). A great deal of recent work has been concerned with feature selection in these applications. Matsui and Konishi (2011); Gertheiss et al. (2013); Fan et al. (2015) study the case where the response is scalar and the features are functional. Chen et al. (2016); Fan and Reimherr (2016); Barber et al. (2017); Parodi et al. (2018); Mirshani and Reimherr (2019) tackle the so called function-on-scalar case, where the response is functional and the features are scalar – focusing on settings in which the number of features is bigger than the number of observations. However, recent developments in optimization have demonstrated that substantial computational gains can still be made when the number of features is massive (e.g $\sim$1e6). In this article, we present Functional Group Elastic Net (fgen), a novel and highly efficient method to solve the function-on-scalar feature selection problem in ultra-high-dimensional settings – where the number of features is indeed massive and much larger than the number of observations. The ability to solve these problems with a lower computational burden is increasingly critical. Given the complex, noisy nature of much contemporary data, changing some aspects of their pre-processing or some of the tuning parameters involved in the analysis can lead to completely different results (Krawczyk and Cano, 2018; Murdoch et al., 2019). For this reason, repeating an analysis multiple

35th Conference on Neural Information Processing Systems (NeurIPS 2021).

times (e.g., with different choices of data preprocessing pipelines, or to tune certain meta-parameters) is paramount to capture significant signals and ensure the stability of outcomes (Yu and Kumbier, 2020). Substantial reductions in computational burden enable such repetition, allowing scientists and practitioners to conduct truly meaningful and reproducible analyses.

Group Elastic Net incorporates the group structure (Yuan and Lin, 2006) and the Elastic Net penalty (Zou and Hastie, 2005) into a penalized regression framework. The former allows one to represent each feature (or component) by a group of variables. The latter induces sparsity and regularizes the estimates. We consider the case where all groups have the same size $k$. The minimization problem is formulated as follows:

$$\min_{B} \ (1/2) \|XB - Y\|_2^2 + \lambda_1 \sum_{i=1}^{p} \|B_i\|_2 + (\lambda_2/2) \sum_{i=1}^{p} \|B_i\|_2^2. \tag{1}$$

Let $p$ be the number of features, $n$ the number of statistical units, and $\| \cdot \|_2$ the $l_2$ norm for matrices, i.e. the Frobenius norm, and vectors. Then, $X \in \mathbb{R}^{n \times p}$ is the design matrix (that we assume to have standardized columns), $Y \in \mathbb{R}^{n \times k}$ the response matrix, and $B \in \mathbb{R}^{p \times k}$ the coefficient matrix. In other words, (1) describes a sparse multi-task model where the response and each of the features are represented by a group of $k$ coefficients (Zhang and Yang, 2018). Throughout this article, we follow the notation in Johnson et al. (2014) and we use the subscripts $i$ and $(i)$ to indicate the $i$-th row and the $i$-th column of a matrix, respectively. Thus, $B_i \in \mathbb{R}^k$ are the coefficient values associated with the $i$-th group, and $X_{(i)} \in \mathbb{R}^n$ are the observed values relative to the $i$-th feature. Before proceeding, note that (1) can be expressed as

$$\min_{B} \big( h(XB) + \pi(B) \big), \tag{P}$$

where $h(XB) = (1/2) \|XB - Y\|_2^2$ is the least-squares loss function and $\pi(B) = \sum_{i=1}^{p} \pi(B_i) = \lambda_1 \sum_{i=1}^{p} \|B_i\|_2 + (\lambda_2/2) \sum_{i=1}^{p} \|B_i\|_2^2$ is the Group Elastic Net penalty function. The first term in $\pi$ is not differentiable and creates sparsity at the group level, i.e, if a component is selected, then all its coefficients are selected and vice-versa. The second term is a Ridge-type penalty which reduces model complexity and tries to control variance inflation due to multicollinearity – feature selection models are indeed known to be less effective and not reliable in scenarios characterized by very high collinearity among features (Katrutsa and Strijov, 2015). $\lambda_1$ and $\lambda_2$ are penalty parameters $> 0$ and control the weight of the two penalties with respect to the least square loss.

To solve (1), we develop a new Semi-smooth Newton Augmented Lagrangian (SsNAL) algorithm. We then extend it to the function-on-scalar regression framework by means of Functional Principal Components (FPC) (James et al., 2000; Chiou et al., 2004; Hall and Hosseini-Nasab, 2006). SsNAL exploits the sparsity induced by the augmented Lagrangian second order information to guarantee a super-linear convergence and greatly reduce the computational cost. This methodology, first introduced by Tomioka and Sugiyama (2009) and Tomioka et al. (2011), has been recently used in several applications, e.g., to regular Lasso (Li et al., 2018), constrained Lasso (Deng and So, 2019), and Elastic-Net (Boschi et al., 2020). However, incorporating the group structure significantly increases the dimension of the problem. Indeed, (1) is not separable and the optimization must be carried out jointly across the coordinates of the outcome. Therefore, considering a new group penalty while preserving the efficiency of the method requires that we carefully redefine a set of all-new mathematical operators and the theory behind them.

We implemented an efficient version of `fgen` in `python` and benchmarked it against the two best Group Elastic Net solvers we found in the literature: the `python package sklearn` (Pedregosa et al., 2011) and the `R package glmnet` (Friedman et al., 2010), which is written in `fortran`. Both of these solvers implement a highly optimized coordinate descent algorithm (Friedman et al., 2010; Breheny and Huang, 2015) and outperform competitors such us FISTA (Beck and Teboulle, 2009; Bonnefoy et al., 2015), ADMM (Deng et al., 2013; Zhu, 2017), and proximal gradient (Chen et al., 2010) by at least one order of magnitude in terms of CPU time. Our simulation results demonstrate that in sparse scenarios `fgen` is at least 3 times faster than `glmnet` and more than 10 times faster than `sklearn`. We also applied `fgen` to the Intervention Nurses Start Infants Growing on Healthy Trajectories (INSIGHT) study (Paul et al., 2014), which investigates risk factors for childhood obesity. Specifically, we examined the association between hundreds of thousand of Single Nucleotide Polymorphisms (SNPs) and growth curves, which represent a functional outcome.

The remainder of the article is organized as follows. In Section 2 we describe the Group Elastic Net problem and introduce some preliminary results. In Section 3 we present our new methodology

and illustrate how to extend it to a function-on-scalar feature selection problem. In Section 4 we investigate the performance of our method on simulated data and apply it to data from INSIGHT. In Section 5 we provide final remarks and discuss future developments. Proofs of theoretical results and additional simulations are included in the Supplement. The `fgen` code is available at `https://github.com/tobiaboschi/fgen`

## 2 Preliminaries

In this section we define the Group Elastic Net problem and we introduce some results related to *Fenchel conjugate functions* and *proximal operators*, which are essential tools in our developments.

### 2.1 Fenchel conjugate function and proximal operator of $\pi(\mathbf{B})$

*Fenchel conjugate functions* (Fenchel, 1949) allow one to more readily define the dual problem (Boyd and Vandenberghe, 2004) of (P), which is called the primal problem. Let $\mathcal{X} \subseteq \mathbb{R}^p$ be a convex set and $f : \mathcal{X} \to \mathbb{R}$. Then, the conjugate function of $f$ is $f^* : \mathcal{X}^* \to \mathbb{R}$ defined as $f^*(z) = \sup_{x \in \mathcal{X}} (\langle z, x \rangle - f(x))$, where $\mathcal{X}^* = \{z : \sup_{x \in \mathcal{X}} (\langle z, x \rangle - f(x)) < \infty\}$. $\langle \cdot, \cdot \rangle$ indicates the inner product. i.e. the dot product. If $\mathcal{X} \subseteq \mathbb{R}^{p \times k}$, i.e. if $z$ is a matrix, the definition is still valid but $\langle \cdot, \cdot \rangle$ is the Frobenius inner product. In our first proposition we provide a closed form solution for the Group Elastic Net penalty conjugate function (see Supplemental Section A.1 for a proof).

**Proposition 1.** *Given $Z \in \mathbb{R}^{p \times k}$, the conjugate function of $\pi$ has the form*

$$\pi^*(Z) = \sum_{i=1}^{p} \pi^*(Z_i) = (2\lambda_2)^{-1} \sum_{i=1}^{p} \left( \left[ \|Z_i\|_2 - \lambda_1 \right]_+ \right)^2, \tag{2}$$

*where $[\ \cdot\ ]_+$ is the positive part operator; $[s]_+ = s$ if $s > 0$ and $0$ otherwise.*

Note that $\pi^*(Z)$ is a continuous differentiable function. This is a more general results than the one presented in Li et al. (2018); Boschi et al. (2020), because we extend the definition of $\pi^*$ to the case where $Z$ is a matrix and not just a vector. In the simple scenario where $k = 1$, i.e. when every group consists of just one variable, we obtain again the conjugate function of the standard Elastic Net penalty. Notably, starting from a non-separable objective function, we derive a $\pi^*$ which does separate. As we will see in Section 3.1, this allows one to induce a new level of sparsity in the Lagrangian problem which is actually key for the massive computational advantage offered by `fgen`.

*Proximal operators* (Rockafellar, 1976a,b) are fundamental in many optimization algorithms. Given a lower semi-continuous convex function $f : \mathbb{R}^p \to \mathbb{R}$, the proximal operator of $f$ at $x$ with parameter $\sigma > 0$ is denoted as $\text{prox}_{\sigma f} : \mathbb{R}^p \to \mathbb{R}^p$ and defined as $\text{prox}_{\sigma f}(x) = \arg \min_t (f(t) + (2\sigma)^{-1} \|t - x\|_2^2)$. If $f : \mathbb{R}^{p \times k} \to \mathbb{R}$, i.e. if $x$ is a matrix, then $\text{prox}_{\sigma f} : \mathbb{R}^{p \times k} \to \mathbb{R}^{p \times k}$ and $\|\cdot\|_2$ is the Frobenius norm. Parikh et al. (2014) and Beck (2017) (Chapter 6) provide numerous examples and properties. Combining their results, one can easily find the form of the proximal operator of $\pi(B)$ provided in our second proposition (see Supplemental Section A.2 for a proof).

**Proposition 2.** *The proximal operator of $\pi(B)$ is:* $\text{prox}_{\sigma\pi}(B) = \left( \text{prox}_{\sigma\pi}(B_1), \ldots, \text{prox}_{\sigma\pi}(B_p) \right)^T$, *where*

$$\text{prox}_{\sigma\pi}(B_i) = (1 + \sigma\lambda_2)^{-1} \left[ 1 - \|B_i\|_2^{-1} \sigma\lambda_1 \right]_+ B_i. \tag{3}$$

Note that $\text{prox}_{\sigma\pi}(B) : \mathbb{R}^{p \times k} \to \mathbb{R}^{p \times k}$. To implement `fgen`, one also needs the proximal operator of $\pi^*$, which can be obtained through the *Moreau decomposition*:

$$x = \text{prox}_{\sigma\pi}(x) + \sigma \, \text{prox}_{\pi^*/\sigma}(x/\sigma) \ , \ \ \sigma > 0. \tag{4}$$

### 2.2 Dual formulation and Augmented Lagrangian

Here we introduce the dual Group Elastic Net problem and its augmented Lagrangian. From Boyd and Vandenberghe (2004), a possible dual formulation of (P) is

$$\min_{V,Z} \left( h^*(V) + \pi^*(Z) \right) \ \text{s.t.} \ \ X^T V + Z = 0 \tag{D}$$

---

**Algorithm 1** Semi-smooth Augmented Lagrangian (SsNAL) method

---

| **Augmented Lagrangian method** | **Semi-smooth Newton method for (7)** |
|---|---|

**Augmented Lagrangian method**

Start from the initial values $V^0, Z^0, B^0, \sigma^0$

**while** not converged **do**

(1) Given $B^k$, find $V^{k+1}$ and $Z^{k+1}$ which approximately solve the inner subproblem

$$\left(V^{k+1}, Z^{k+1}\right) \approx \arg\min_{V,Z} \mathcal{L}_\sigma\left(V, Z \mid B^k\right) \quad (7)$$

(2) Update the Lagrangian multiplier $B$ and the parameter $\sigma$:

$$B^{k+1} = B^k - \sigma_k\left(X^T V^{k+1} + Z^{k+1}\right) \qquad (8)$$

$$\sigma^{k+1} \uparrow \sigma^\infty \leq \infty$$

**end while**

**Semi-smooth Newton method for (7)**

To solve (7) and find $\left(V^{k+1}, Z^{k+1}\right)$:

**while** not converged **do**

(1) Find the descent direction $D^j$ solving exactly or by *conjugate gradient* the linear system

$$\partial^2\psi(V^j)\,\mathrm{vec}(D^j) = -\,\mathrm{vec}\left(\nabla\psi(V^j)\right) \quad (9)$$

(2) *Line search* (Li et al., 2018): choose $\mu \in (0, 1/2)$ and reduce the step size $s^j$ until

$$\psi\left(V^j + s^j D^j\right) \leq \psi(V^j) + \mu s^j\left\langle \nabla\psi(V^j), D^j \right\rangle$$

(3) Update $V$: $V^{j+1} = V^j + s^j D^j$

(4) Update $Z$: $Z^{j+1} = \mathrm{prox}_{\frac{\pi^*}{\sigma}}\left(\frac{B^k}{\sigma^k} - X^T V^{j+1}\right)$

**end while**

---

where $V \in \mathbb{R}^{n \times k}$ and $Z \in \mathbb{R}^{p \times k}$ are the dual variables matrices. In particular $V_i, Z_i \in \mathbb{R}^k$ are the dual variables associated with the $i$-th group. $h^*$ and $\pi^*$ are the Fenchel conjugate functions of $h$ and $\pi$, respectively. Specifically, $h^*(V) = (1/2)\|V\|_2^2 + \langle Y, V \rangle$ (Dünner et al., 2016) and $\pi^*(Z)$ is given in Proposition 1. We can now define the augmented Lagrangian function and the *Karush-Kuhn-Tucker* (KKT) system associated with (D). The augmented Lagrangian is given by

$$\mathcal{L}_\sigma(V, Z, B) = h^*(V) + \pi^*(Z) - \sum_{i=1}^p \left\langle B_i, V^T X_{(i)} + Z_i \right\rangle + (\sigma/2)\sum_{i=1}^p \left\|V^T X_{(i)} + Z_i\right\|_2^2, \quad (5)$$

where $\sigma > 0$. $B$ is both the primal variable and the Lagrangian multiplier which penalizes the constraints' violations. The KKT system is given by the following three equations:

$$\nabla h^*(V) - XB = 0, \qquad 0 = \nabla\pi^*(Z) - B = 0, \qquad X^T V + Z = 0. \quad (6)$$

Note that $\nabla h^*(V) = V + Y$. A closed form of $\nabla\pi^*(Z)$ is not essential for our SsNAL method. The KKT equations will be useful to determine the convergence of our algorithm, since the set $(V^\star, Z^\star, B^\star)$ solves the KKT (6) if and only if $(V^\star, Z^\star)$ and $B^\star$ are the optimal solutions of (D) and (P), respectively (Boyd and Vandenberghe, 2004).

# 3 Methodology

In this section we present our new methodology. First, we introduce a SsNAL algorithm to solve the Group Elastic Net problem. Next, we illustrate how to extend it to the function-on-scalar regression framework. Finally, we describe how to implement a solution path over different values of $\lambda_1$.

## 3.1 SsNAL method

The SsNAL method is summarized in **Algorithm 1**. It consists of an *Augmented Lagrangian method* characterized by an inner subproblem. The subproblem is solved with a *Semi-smooth Newton method* which exploits the sparsity of the augmented Lagrangian second order information and greatly reduces computational costs. We now provide the details of its implementation and some important theoretical results. From Rockafellar (1976a), one can find the optimal solution of (D) by solving the Augmented Lagrangian method described in **Algorithm 1**. The essential part of the algorithm is the subproblem (7). As described in Li et al. (2018), an approximate solution $\left(\bar{V}, \bar{Z}\right)$ for a given $B$ can be found as

$$\bar{V} = \arg\min_V \mathcal{L}_\sigma\left(V \mid \bar{Z}, B\right), \quad \bar{Z} = \arg\min_Z \mathcal{L}_\sigma\left(Z \mid \bar{V}, B\right). \quad (10)$$

With a slight abuse of notation, we indicate by $L_\sigma(V|Z, B)$ the function $L_\sigma(V, Z, B)$ where the parameter Z and B are fixed. Similarly for $L_\sigma(Z|V, B)$. Our third proposition provides explicit forms for $\mathcal{L}_\sigma\left(V \mid \bar{Z}, B\right)$ and $\bar{Z}$ (see Supplemental Section A.3 for a proof).

**Proposition 3.** *Define* $\psi(V) := \mathcal{L}_\sigma\left(V \mid \bar{Z}, B\right)$. *Then, for the Group Elastic Net problem we have*

$$(a)\ \psi(V) = h^*(V) + \frac{1 + \sigma\lambda_2}{2\sigma} \sum_{i=1}^p \left\| \operatorname{prox}_{\sigma\pi}\left(B_i - \sigma V^T X_{(i)}\right) \right\|_2^2 - \frac{1}{2\sigma} \sum_{i=1}^p \|B_i\|_2^2 \quad (11)$$

$$(b)\ \bar{Z} = \operatorname{prox}_{\pi^*/\sigma}\left(B/\sigma - X^T\bar{V}\right),$$

*where* $\operatorname{prox}_{\pi^*/\sigma}\left(B/\sigma - X^T\bar{V}\right) = \left( \operatorname{prox}_{\pi^*/\sigma}\left(B_1/\sigma - \bar{V}^T X_{(1)}\right), \cdots, \operatorname{prox}_{\pi^*/\sigma}\left(B_p/\sigma - \bar{V}^T X_{(p)}\right) \right)^T.$

$\bar{Z}$ has a closed form. To find $\bar{V}$ one has to minimize $\psi$ or, equivalently, find the solution of $\nabla\psi = 0$. Note that $\psi$ is continuous and differentiable, and thus $\nabla\psi$ is well defined.

To solve the subproblem (7), we propose the Semi-smooth Newton method in **Algorithm 1**. $V$ and $Z$ are updated iteratively – $Z$ according to the rule in Proposition 3, and $V$ by minimizing $\psi$ through one Newton step. The main computational cost is solving the linear system (9). This leads to our next crucial result (see Supplemental Section A.4 for a proof).

**Theorem 1.** *Let* $T = B - \sigma X^T V$, $\hat{X} = X \otimes I_k$ *(the $nk \times pk$ Kronecker product between $X$ and the $k \times k$ identity matrix), $\hat{\partial}^2\psi$ be the generalized Hessian of $\psi$, and $\partial\operatorname{prox}_{\sigma\pi}$ be the Clarke sub-differential of $\operatorname{prox}_{\sigma\pi}$ (Clarke, 1990). Then we have*

$$(i)\nabla\psi(V) = V + Y - X\operatorname{prox}_{\sigma\pi}(T) \quad (ii)\hat{\partial}^2\psi(V) = I_{nk} + \sigma\hat{X}\partial\operatorname{prox}_{\sigma\pi}(T)\hat{X}^T \quad (12)$$

*Moreover, let $Q \in \mathbb{R}^{pk \times pk}$ be the block-diagonal matrix* $Q = \begin{bmatrix} P_1 & & \\ & \ddots & \\ & & P_p \end{bmatrix}$, *where each $P_i$ is a squared $k \times k$ matrix defined as*

$$(iii)\ P_i = \begin{cases} (1 + \sigma\lambda_2)^{-1}\left(1 - \|T_i\|_2^{-1}\sigma\lambda_1\right)I_k + \|T_i\|_2^{-3}\sigma\lambda_1 T_i T_i^T & \|T_i\|_2 > \sigma\lambda_1 \\ 0 & o.w. \end{cases}. \quad (13)$$

*Then $Q \in \partial\operatorname{prox}_{\sigma\pi}(T)$ and $\partial^2\psi(V)\operatorname{vec}(D) = (I_{nk} + \sigma\hat{X}Q\hat{X}^T)\operatorname{vec}(D)$ for every $D \in \mathbb{R}^{n \times k}$ in the domain of $V$ – where $\operatorname{vec}(D) \in \mathbb{R}^{nk}$ is obtained by stacking all the columns of $D$.*

Note that, while in Li et al. (2018); Deng and So (2019); Boschi et al. (2020) $\nabla\psi$ and $\partial^2\psi$ are a vector and a matrix, respectively, here the dimensions of these operators increase due to the group nature of the problem. In particular, $\nabla\psi$ becomes a matrix and $\hat{\partial}^2\psi$ a higher order tensor – which we express as an $nk \times nk$ matrix by stacking its dimensions. Moreover, $Q$ is not a simple diagonal matrix as in the previous SsNAL algorithms, but is now characterized by blocks associated to the different groups of variables.

Theorem 1 is critical for preserving the efficiency of `fgen`, while integrating groups into the problem. First, it states that solving (9) is equivalent to solving $\left(I_{nk} + \sigma\hat{X}Q\hat{X}^T\right)\operatorname{vec}(D) = -\operatorname{vec}\left(\nabla\psi(V)\right)$. Second, the form of $Q$ still allows one to induce sparsity in the linear system and drastically reduce the computational cost. Indeed, let $\mathcal{J} = \left\{j : \|T_j\|_2 \geq \sigma\lambda_1\right\}$ and let $r = |\mathcal{J}|$ be the cardinality of $\mathcal{J}$. Then the linear system (9) is equivalent to

$$\left(I_{nk} + \sigma\hat{X}_{\mathcal{J}}Q_{\mathcal{J}}\hat{X}_{\mathcal{J}}^T\right)\operatorname{vec}(D) = -\operatorname{vec}\left(\nabla\psi(V)\right). \quad (14)$$

Here, $\hat{X}_{\mathcal{J}} \in \mathbb{R}^{nk \times rk}$ is defined as $\hat{X}_{\mathcal{J}} = X_{\mathcal{J}} \otimes I_k$, with $X_{\mathcal{J}} \in \mathbb{R}^{n \times r}$ being the sub-matrix of $X$ restricted to the columns in $\mathcal{J}$. In addition, $Q_{\mathcal{J}} \in \mathbb{R}^{rk \times rk}$ is the block-diagonal matrix formed by all the $P_i$ such that $i \in \mathcal{J}$. Using the *Cholesky factorization* (the generalized Hessian is positive semidefinite) the total cost of solving the linear system reduces from $\mathcal{O}\left(nk^3(n^2 + p^2 + np)\right)$ to $\mathcal{O}\left(nk^3(n^2 + r^2 + nr)\right)$. This includes computing $\hat{X}_{\mathcal{J}}Q_{\mathcal{J}}\hat{X}_{\mathcal{J}}^T$, which is $\mathcal{O}\left(nrk^3(n + r)\right)$, and the Cholesky factorization, which is $\mathcal{O}(n^3k^3)$. Because of the sparsity induced by the Group Elastic Net penalty, $r$ is usually much smaller than $p$ – implying a substantial computational gain. Even when $p$ is very large ($\sim 10^6$), one can still solve the linear system efficiently, as long as the dimension $k$ of each group is relatively small ($< 10^2$). Furthermore, if $r < n$, which is often the case when

---

**Algorithm 2** Functional Group Elastic Net method

---

**(1)** Perform FPC of $\mathcal{B}$ and find the first $k$ basis components $(\gamma_i, \dots \gamma_k)$ with their eigenvalues $(\rho_1, \dots \rho_k)$

**(2)** Find the first $k$ FPC scores for each response function $\mathcal{Y}_i$: $Y_i = (\langle \mathcal{Y}_i, \gamma_1 \rangle_{\mathbb{L}^2}, \dots, \langle \mathcal{Y}_i, \gamma_k \rangle_{\mathbb{L}^2})$

**(3)** Using $Y$ as response matrix, apply SsNAL to solve (1) and find the coefficient scores estimates $B$.

**(4)** Project $B$ into the FPC basis to find the coefficient curve estimates: $\mathcal{B}_i = \sum_{j=1}^{k} B_{(i,j)} \gamma_j$

---

the solution of the Group Elastic Net problem is sparse, one can factorize an $rk \times rk$ (instead of $nk \times nk$) matrix using the *Sherman-Morrison-Woodbury* formula (Van Loan and Golub, 1983):

$$\left(I_{nk} + \sigma \hat{X}_{\mathcal{J}} Q_{\mathcal{J}} \hat{X}_{\mathcal{J}}^T\right)^{-1} = I_{nk} - \hat{X}_{\mathcal{J}} \left((\sigma Q_{\mathcal{J}})^{-1} + \hat{X}_{\mathcal{J}}^T \hat{X}_{\mathcal{J}}\right)^{-1} \hat{X}_{\mathcal{J}}^T. \tag{15}$$

The total cost is further reduced from $\mathcal{O}\left(nk^3(n^2 + r^2 + nr)\right)$ to $\mathcal{O}\left(rk^3(n^2 + r^2 + nr + 1)\right)$, including the computation of $Q_{\mathcal{J}}^{-1}$ which can be done with a cost of $\mathcal{O}(rk^3)$ by inverting each one of the $P_i$ blocks independently. Finally, if in the first iterations of the algorithm $n$ and $r$ are both larger than $10^4$, one can solve (9) approximately using the *conjugate gradient method* (Polyak, 1969).

To determine the convergence of the Augmented Lagrangian and the Semi-smooth Newton methods, we check the residuals of the third and first KKT in (6), respectively, i.e. :

$$\text{res(kkt}_3) = \frac{\sum_{i=1}^{p} \left\|V^T X_{(i)} + Z_i\right\|_2}{1 + \sum_{i=1}^{m} \|V_i\|_2 + \sum_{i=1}^{p} \|Z_i\|_2}, \quad \text{res(kkt}_1) = \frac{\sum_{i=1}^{n} \|V_i + Y_i - X_i B\|_2}{1 + \sum_{i=1}^{n} \|Y_i\|_2}. \tag{16}$$

Taking the $l2$-norm of the KKT residuals, normalizing them and using them to assess convergence is a common procedure in the literature (Li et al., 2018; Deng and So, 2019). Both methods have a super-linear convergence rate. Accordingly, the convergence rate of the entire algorithm, which is the sum of the convergence rate of the two sub-problems (Tomioka and Sugiyama, 2009), is still super-linear. Thus, as we show in Section 4, `fgen` typically converges in very few iterations. The convergence analysis here follows directly from that in Tomioka et al. (2011) and Boschi et al. (2020), where SsNAL convergence is proved for the standard Elastic Net. The proof leverages results in Rockafellar (1976a,b); Luque (1984); Li et al. (2018) and the fact that $\pi^*$ is a continuous differentiable function (which is also true for the Group Elastic Net).

## 3.2 Extension to function-on-scalar regression

We now extend `fgen` to the function-on-scalar features selection problem. In function-on-scalar regression, a functional response is regressed on a set of scalar predictors. Assuming the response belongs to the Hilbert Space $\mathbb{L}^2([a, b])$, the optimization problem (1) becomes

$$\min_{\mathcal{B}} \ (1/2)\|X\mathcal{B} - \mathcal{Y}\|_{\mathbb{L}^2}^2 + \lambda_1 \sum_{i=1}^{p} \|\mathcal{B}_i\|_{\mathbb{L}^2} + (\lambda_2/2) \sum_{i=1}^{p} \|\mathcal{B}_i\|_{\mathbb{L}^2}^2 \tag{17}$$

where $\mathcal{Y}$ and $\mathcal{B}$ are functional objects with $n$ and $p$ rows, respectively. Each row $\mathcal{Y}_i$ is a response function and each row $\mathcal{B}_i$ is a coefficient function. The squared $\mathbb{L}^2$-norm of a function $f$ is $\|f\|_{\mathbb{L}^2}^2 = \langle f, f \rangle_{\mathbb{L}^2}$, where the inner product between two functions $f$ and $g$ is $\langle f, g \rangle_{\mathbb{L}^2} = \int_a^b fg$.

Applying SsNAL directly to (17) is not straightforward and would substantially hinder its efficiency. First, the definition of conjugate functions and proximal operators in functional spaces would require a new theoretical background. Second, and perhaps most important from a practical standpoint, computing integrals is much more expensive than computing euclidean norms. For these reasons, in **Algorithm 2** we take advantage of *Functional Principal Components* (FPC) (Horváth and Kokoszka, 2012; Kokoszka and Reimherr, 2017) to solve an optimization problem of the same type as (1), which is in fact a very close approximation to (17). In particular, we build a response matrix $Y$, where each group $i$ is formed by the first $k$ FPC scores of the the function $\mathcal{Y}_i$ (Fan and Reimherr, 2016). The level of approximation of `fgen` thus depends on the number of FPC scores $k$, i.e the dimension of each group. Indeed, given a function $f$ and its FPC basis $\{\gamma_i\}_{i=1}^{\infty}$, we have

$$\|f\|_{\mathbb{L}^2} = \sum_{j=1}^{\infty} \|\langle f, \gamma_j \rangle_{\mathbb{L}^2}\|_2 \ . \tag{18}$$

This property – which is true for every orthonormal basis system – plays a crucial role in the extension of our SsNAL approach to the function-on-scalar regression, since it allows one to approximate the $\mathbb{L}^2$ function norm with the standard $l_2$ matrix norm. Consequently, one can use the FPC scores to construct the response matrix $Y$ and the coefficients matrix $B$ in (1) starting from the response functions $\mathcal{Y}$ and the coefficient functions $\mathcal{B}$. The number of selected FPC scores determines the dimension $k$ of each group in the Group Elastic Net problem. In many applications just a few FPC scores allow one to obtain a very close approximation of the original functions. Indeed, among the many orthonormal bases one could envision, FPC has the advantage of being the most parsimonious allowing one to reconstruct the response curves using fewer coefficients than any other orthonormal basis. In scenarios investigated by simulation in Section 4, $k = 5$ is sufficient to capture more than the 99% of the $\mathbb{L}^2$-norm. This produces an almost perfect approximation of (17) while fully preserving `fgen` efficiency.

### 3.3 Solution path implementation

To evaluate different values of the penalty parameter $\lambda_1$, we implement an efficient solution path search. We compute the solution for a decreasing sequence sequence of $\lambda_1$, starting from $\lambda^{max} = \max_i \|(X_i)^T Y\|$ which selects 0 active features. When we move to the next $\lambda_1$ value, we use the solution obtained at the previous value for initialization (*warm start*). The two consecutive solutions tend to be close, and `fgen` converges in very few iterations – usually just one. We also allow the user to specify a maximum number of selected features; when this number is reached the path search is stopped, further reducing computation.

To guide the choice of $(\lambda_1, \lambda_2)$ we propose two quantitative criteria: *k-fold Cross Validation* ($cv$) and an *Extended Bayesian Information Criterion* ($e\text{-}bic$) (Chen and Chen, 2012), which modifies the standard BIC to also include the number of features $p$. In symbols, we have

$$\text{e-bic}(B) = k \log\left(\text{rss}(B)/(nk)\right) + (k\nu)\left(\log(nk) + \log p\right)/n \qquad (19)$$

where $\text{rss}(B)$ is the residual sum of squares associated with the solution $B$, and $\nu$ are the Group Elastic Net degrees of freedom. From Tibshirani et al. (2012), $\nu = \text{tr}\left(X_{\mathcal{J}}\left(X_{\mathcal{J}}^T X_{\mathcal{J}} + \lambda_2 I_r\right)^{-1} X_{\mathcal{J}}^T\right)$. Note that $cv$ can be very computationally expensive because it requires to run `fgen` multiple times for each value of $\lambda_1$ and $\lambda_2$ under consideration. In contrast, $e\text{-}bic$ can be computed directly from the original solution. Before evaluating both criteria, we de-bias the `fgen` estimates following the approach suggested by Belloni et al. (2014); Zhao et al. (2017). First, we run `fgen`, then, we fit a standard least squares on the selected features. In the next section, following standard practice in the literature – e.g., Friedman et al. (2010); Pedregosa et al. (2011) – we rewrite $\lambda_1$ and $\lambda_2$ as $\lambda_1 = c_\lambda \lambda^{max}$ and $\lambda_2 = (1-\alpha)c_\lambda \lambda^{max}$, with $c_\lambda \in (0, 1]$ and $\alpha \in (0, 1)$. $c_\lambda$ determines the reduction with respect of $\lambda^{max}$, $\alpha$ controls the relative weight of the two penalties.

## 4 Simulation study and INSIGHT data

In this section we use synthetic data to illustrate the computational efficiency of `fgen`, and apply our new method to a Genome Wide Association Study (GWAS) on childhood obesity. In the simulations, we benchmark `fgen` against the two best Group Elastic Net solvers we found in the literature: the `python package sklearn` and `R package glmnet`, which is written in `fortran`. Other functional-on-scalar feature selection methods, such as the ones proposed by Barber et al. (2017); Parodi et al. (2018); Mirshani and Reimherr (2019), have a computational burden more than two order of magnitudes larger than `fgen` and could not complete instances with $p > 10^4$.

### 4.1 Simulation results

We generate synthetic data as follows. The entries of the design matrix $X \in \mathbb{R}^{n \times p}$ are each drawn independently from a standard normal distribution. The response curves are created as $\mathcal{Y} = X\mathcal{B} + \epsilon$. $\mathcal{B}$ contains $p_0$ non-zero curves. These and the errors $\epsilon$ are generated from a 0 mean Gaussian process with a Matérn covariance function (Cressie and Huang, 1999) of the form

$$C(t, s) = \omega^2 \left(\Gamma(\nu)2^{\nu-1}\right)^{-1} \left((l)^{-1}(2\nu)^{1/2}|t-s|\right)^\nu K_\nu\left((l)^{-1}(2\nu)^{1/2}|t-s|\right), \qquad (20)$$

where $K_\nu$ is a modified Bessel function. In particular, we set the point-wise variance $\omega^2 = 1$ and the range $l = 0.25$ (this determines how fast the curves dependency decays). The smooth parameter $\nu$

Table 1: $a$, $b$ and $c$ report CPU time in seconds for `fgen`, `sklearn` and `glmnet`, respectively. For `fgen` we also report the number of iterations in parenthesis. $r$ is the number of selected features, $l$ is the range parameter of the Matern process used to generate the coefficients.

| $\alpha=0.8,$  $l=0.25$ | | | n=500 | | | | n=1000 | | | | n=5000 | | | |
|---|---|---|---|---|---|---|---|---|---|---|---|---|---|---|
| $p; p_0$ | k | $c_\lambda$ | r | a | b | c | r | a | b | c | r | a | b | c |
| $2(10^4); 10$ | 5 | 0.8 | 2 | **0.1**(3) | 0.5 | 0.3 | 1 | **0.1**(2) | 0.9 | 0.6 | 1 | **0.5**(2) | 9.1 | 3.2 |
| | | 0.4 | 10 | **0.2**(4) | 0.5 | 0.3 | 4 | **0.3**(3) | 1.1 | 0.6 | 6 | **1.1**(3) | 10.9 | 3.1 |
| | | 0.2 | 21 | **0.4**(4) | 0.5 | **0.4** | 8 | **0.3**(4) | 1.1 | 0.7 | 10 | **1.2**(3) | 10.4 | 3.4 |
| | 10 | 0.8 | 2 | **0.2**(3) | 0.9 | 0.5 | 1 | **0.2**(2) | 1.9 | 1.2 | 1 | **1.3**(2) | 18.8 | 5.4 |
| | | 0.4 | 10 | **0.3**(4) | 0.8 | 0.5 | 4 | **0.4**(3) | 2.1 | 1.0 | 6 | **2.0**(3) | 22.3 | 4.9 |
| | | 0.2 | 21 | 0.9(4) | 0.9 | **0.5** | 8 | **0.5**(4) | 2.2 | 1.2 | 10 | **1.9**(3) | 23.2 | 5.3 |
| $10^5; 10^2$ | 5 | 0.8 | 5 | **0.3**(2) | 4.5 | 1.5 | 4 | **0.7**(2) | 9.9 | 3.3 | 6 | **2.6**(2) | 97.9 | 17.2 |
| | | 0.6 | 29 | **0.6**(2) | 4.7 | 1.6 | 17 | **0.7**(2) | 9.6 | 3.3 | 12 | **3.0**(2) | 85.7 | 19.8 |
| | | 0.4 | 486 | 13.7(3) | 4.8 | **0.2** | 53 | **1.4**(2) | 10.2 | 3.1 | 42 | **4.1**(2) | 99.7 | 17.8 |
| | 10 | 0.8 | 5 | **0.6**(2) | 10.2 | 2.6 | 4 | **1.4**(2) | 22.6 | 5.2 | 6 | **3.5**(2) | 175.2 | 27.9 |
| | | 0.6 | 29 | **0.9**(2) | 10 | 2.4 | 17 | **1.5**(2) | 19.6 | 4.8 | 12 | **3.8**(2) | 183.7 | 27.6 |
| | | 0.4 | 486 | 79.2(3) | 9.1 | **2.4** | 53 | **1.8**(2) | 20.2 | 5.3 | 42 | **6.7**(2) | 190.7 | 28.0 |

| $\boldsymbol{\alpha=0.5,}$  $l=0.25$ | | | n=500 | | | | n=1000 | | | | n=5000 | | | |
|---|---|---|---|---|---|---|---|---|---|---|---|---|---|---|
| $p; p_0$ | k | $c_\lambda$ | r | a | b | c | r | a | b | c | r | a | b | c |
| $2(10^4); 10$ | 10 | 0.8 | 2 | **0.1**(2) | 1.0 | 0.6 | 1 | **0.2**(2) | 1.8 | 1.3 | 1 | **0.9**(2) | 18.3 | 5.2 |
| | | 0.4 | 10 | **0.2**(3) | 0.9 | 0.6 | 3 | **0.2**(2) | 2.5 | 0.9 | 7 | **0.9**(2) | 20.6 | 5.3 |
| | | 0.2 | 73 | 1.1(3) | 0.9 | **0.5** | 8 | **0.4**(3) | 2.0 | 1.3 | 10 | **1.9**(3) | 22.7 | 5.2 |

| $\alpha=0.8,$  $\boldsymbol{l=0.10}$ | | | n=500 | | | | n=1000 | | | | n=5000 | | | |
|---|---|---|---|---|---|---|---|---|---|---|---|---|---|---|
| $p; p_0$ | k | $c_\lambda$ | r | a | b | c | r | a | b | c | r | a | b | c |
| $2(10^4); 10$ | 10 | 0.8 | 3 | **0.2**(3) | 0.9 | 0.6 | 3 | **0.2**(2) | 1.8 | 1.3 | 1 | **0.9**(2) | 19.7 | 5.1 |
| | | 0.4 | 10 | **0.3**(4) | 1.2 | 0.6 | 7 | **0.4**(3) | 2.6 | 1.1 | 10 | **1.9**(3) | 19.8 | 5.2 |
| | | 0.2 | 20 | 1.5(4) | 1.2 | **0.6** | 10 | **0.5**(4) | 2.4 | 1.2 | 10 | **2.0**(3) | 22.2 | 4.0 |

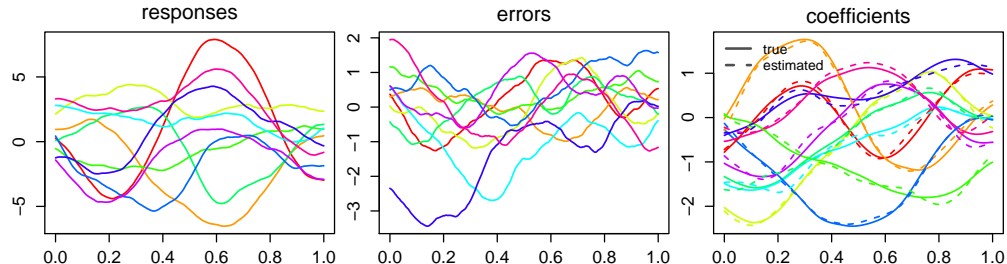

Figure 1: Curves related to a simulation scenario described in Table 1 (second row, $n = 500$). The left and center panels display a sample of 10 response curves and 10 linear model errors terms, respectively. The right panel depicts the true non-zeros coefficients curves (dashed lines) and the `fgen` de-biased estimates (solid lines).

is equal to $3.5$ for $\mathcal{B}$ and to $1.5$ for $\epsilon$, i.e. the errors are rougher than the coefficients. Each curve is sampled at 1000 evenly spaced points between 0 and 1. Figure 1 shows instances of the response and error curves, $\mathcal{Y}$ and $\epsilon$, and the true non-zero coefficient curves in $\mathcal{B}$ along with their de-biased estimates produced by `fgen` (the underlying simulation parameters are those in Table 1, second row, $n = 500$). In all scenarios, `fgen` is run with both the tolerances in (16) set to $10^{-6}$ (we set the same tolerance for `sklearn` and `glmnet`) and $\mu$ in (12) set to 0.2. We start from $\sigma^0 = p_0/p$ and increase it by a factor of 5 every iteration. If we start from smaller values of $\sigma$, the algorithm needs more iterations to converge, while if $\sigma^0$ is too large, `fgen` does not converge to the optimal solution. We set $\lambda_1 = c_\lambda \lambda^{max}$ and $\lambda_2 = (1 - \alpha)c_\lambda \lambda^{max}$, where $c_\lambda \in (0,1]$, $\alpha \in (0,1)$, and $\lambda^{max} = \max_i \|(X_i)^T Y\|$. Note that for `glmnet` and `sklearn` we need to divide $\lambda^{max}$ by $n$ since both solvers divide the least squares loss in (1) by the number of observations.

Table 1 reports CPU times for `fgen`, `sklearn` and `glmnet` under different simulation settings. `fgen` is the fastest solver in almost every instance. When both $n$ and $p$ are large and the solution is sparse, `fgen` is approximately 6 times faster than `glmnet` and more than 30 times faster than `sklearn`. Note

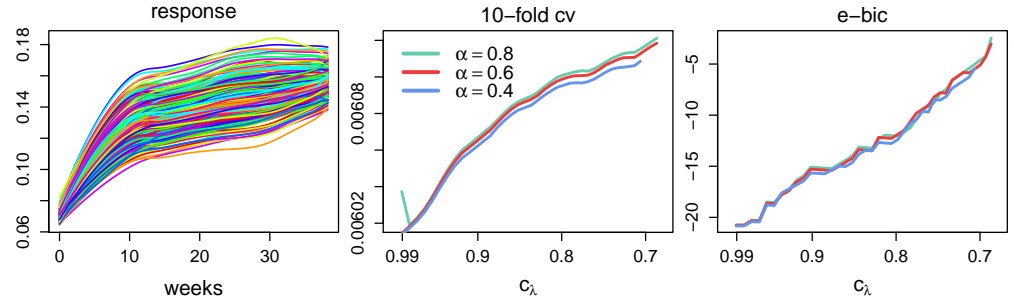

Figure 2: Plots related to the INSIGHT study. The left panel displays the *growth curves*. The center and right panels depict values of the 10-fold *cv* Mean Squared Error and the *e-bic*, respectively, against $c_\lambda$ These are obtained from `fgen` run with 3 different values of $\alpha$; 0.8 (green line), 0.6 (red line), 0.4 (blue line). `fgen` estimates are de-biased prior to computing both criteria.

that the super-linear convergence rate allows `fgen` to converge in very few iterations (no more than 4 in all cases). The CPU time increases with $k$ for all solvers. However, $k = 5$ already captures more than the $99\%$ of the $\mathbb{L}^2$-norm in all the scenarios considered. If we decrease $\alpha$ from $0.8$ to $0.5$, `fgen` need even fewer iterations to converge, increasing its computational gain with respect to the competitors. Considering rougher coefficients (created with a Matern process with range parameter $l = 0.1$) does not affect the relative performance of the algorithms. The instance with an active set of $486$ features is the only one where `fgen` performs worse than its competitors. As expected, in the presence of non-sparse solutions `fgen` looses some of its efficiency. However, to tune the penalty parameters in practice, one evaluates a sequence of $c_\lambda$ values starting from very sparse solutions. In the first steps of the solution path, `fgen` exploits sparsity and is very efficient. In the following steps, it still converges very quickly thanks to the *warm-start* approach described in Section 3.3. In Supplemental Table B.2 we compare the solution path computing time. `fgen` outperforms the other solvers in every scenario, being approximately 2 times faster than `glmnet` and from 10 to more than 30 times faster than `sklearn`. Finally, to gauge uncertainty in our CPU time evaluations, we replicated a subset of the instances explored in Table 1 20 independent times. Mean CPU times and standard errors over such replicates are reported in Supplemental Table B.1. Results agree with those obtained considering just one replication. Furthermore, one can notice that `fgen` has also a smaller variability in CPU time when $n = 5000$.

Taking into account that `glmnet` (written in `fortran`) and `sklearn` are highly optimized packages, the results above provide strong evidence in support of our method. We also tracked prediction performance for all methods and in all simulation settings considered, but we do not report them here since all three solvers solve the same convex minimization problem and therefore converged to the same solution in all settings.

## 4.2 INSIGHT study

Here, we apply `fgen` to data from the Intervention Nurses Start Infants Growing on Healthy Trajectories (INSIGHT) study (Paul et al., 2014). In particular, we focus on data collected to investigate genetic variants that may affect the risk of childhood obesity. As the prevalence of obesity increases also among children, examining possible causes and risk factors has become an essential public health concern. INSIGHT provides genome-wide Single Nucleotide Polymorphisms (SNPs) information for a cohort of very young children, along with longitudinal information on their growth. Selecting SNPs that may affect growth is thus a GWAS (a Genome-Wide Association Study) – where the outcome is a *growth curve*. In recent years, many GWASs have identified SNPs strongly associated with obesity phenotypes (Locke et al., 2015). Before proceeding with the analysis, we point out that due to high feature collinearity, low signal-to-noise ratio, and ultra-high dimensionality, GWAS data are very hard to examine and users should be very careful in interpreting results; e.g., selected SNPs may just be proxies for other causal SNPs in their vicinity. While being well aware of all its complexities and potential pitfalls, our main aim in presenting a GWAS analysis is to show the efficiency and the broad applicability of `fgen`. Our functional outcome captures the evolution of *weight/height* ratios (Daniels et al., 2015) measured at birth and at 4, 16, 28, and 40 weeks for a total of $n = 210$ children. The growth curves – shown in the left panel of Figure 2 – are fitted as in Craig et al. (2019) using

Principal Analysis by Conditional Estimation (Chen et al., 2017). Building a smooth curve for each child allows one to capture information along the entire time domain and at the same time to de-noise and mitigate the effect of outlying/anomalous raw measurements. The SNPs collected in INSIGHT are available upon request at dbGaP using the access number phs001498.v1.p1. The growth curves are based on privacy protected data and cannot be made publicly available.

Craig et al. (2019) used `flame` (Parodi et al., 2018) to solve the function-on-scalar feature selection problem. To do so, they had to reduce the analysis from $p = 342325$ to 10000 SNPs with various preliminary screening steps. The computational efficiency of `fgen` allows us to inspect all the 342325 SNPs simultaneously. The center and left panels of Figure 2 display 10-fold $cv$ and $e\text{-}bic$ for different values of $\alpha$ and $c_\lambda$. Both criteria identify just *one* ($c_\lambda = 0.99$) dominant SNP, $rs79187646$. Without drawing any strong domain conclusion, we remark that the selected SNP appears to be relevant in the literature. Notably, this is the same SNP selected by Boschi et al. (2020), where the same SNP data were associated to BMI at age 3 – a scalar response. Also notably, this dominant SNP was not among the SNPs identified by Craig et al. (2019). Based on the U.S. National Library of Medicine, $rs79187646$ is located in NTM, a well known gene. According to the NHGRI-EBI GWAS Catalog, many GWASs, including two recent studies (Kichaev et al., 2019; Pulit et al., 2019), have connected NTM to body mass, food addiction, intake of sweet substances and other obesity-related traits.

## 5 Conclusions

In this article we proposed a new Function Group Elastic Net method (`fgen`) to solve the function-on-scalar feature selection problem. Our proposal starts with the development of a novel, highly-efficient SsNAL algorithm to solve the Group Elastic Net – which is then extended to the function-on-scalar regression framework using a Functional Principal Components representation. Though we could rely on critical prior results (Tomioka and Sugiyama, 2009; Li et al., 2018; Boschi et al., 2020), in order to integrate the group structure into SsNAL, we had to tackle more complex mathematical operators and redefine the theoretical foundation. Our simulations show a substantial reduction in CPU time with respect to the best existing Group Elastic Net solvers. Finally, we applied `fgen` to a GWAS study detecting a SNP that may affect obesity risk in children.

The current version of `fgen` is limited to the case where each group has the same size $k$ and to the functional-on-scalar feature selection problem. In the future, we plan to further extend our work investigating more complex optimization problems (e.g., allowing each group to have a different size) and functional regression frameworks. In particular, we aim to adapt our methodology to the function-on-function feature selection scenario, where both the response and the predictors can be represented as functional curves.

### Acknowledgments and Disclosure of Funding

We thank Kateryna Makova and her laboratory at Penn State for access to the INSIGHT data, Ana Kenney for help with the data and useful discussions. The work of Tobia Boschi and Francesca Chiaromonte was partially supported by the Huck Institutes of the Life Sciences at Penn State, the work of Matthew Reimherr was partially supported by the Grant NSF SES-1853209.

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
