# Supplement

## A. Proofs

In this section we provide detailed proofs of our theoretical results.

### A.1. Proof of Proposition 1

Note that $\pi(B) = \sum_{i=1}^{p} \pi(B_i)$ is a separable sum. From Boyd and Vandenberghe (2004) we have

$$\pi^*(Z) = \sum_{i=1}^{p} \pi^*(Z_i) . \tag{A1}$$

To compute $\pi^*(Z_i)$, we use a result from Touchette (2005). Define $g(b) = (Z_i)^T b - \pi(b) = (Z_i)^T b - \lambda_1 \|b\|_2 - (\lambda_2/2) \|b\|_2^2$, and let $b^\star = \arg\max_{b \in \mathbb{R}^p} g(b)$. Then

$$\pi^*(Z_i) = g(b^*) . \tag{A2}$$

To find $b^*$ we have to solve $\nabla g(b) = 0$, where $\nabla g(b) = Z_i - \lambda_2 - \lambda_1 \begin{cases} \|b\|_2^{-1} b & \|b\|_2 \neq 0 \\ \{b \,:\, \|b\|_2 \leq 1\} & o.w. \end{cases}$. Consider the case where $\|b\|_2 \neq 0$ and set $\nabla g(b) = 0$. We get

$$Z_i = \left(\lambda_2 + \|b\|_2^{-1} \lambda_1\right) x . \tag{A3}$$

To solve for $b$ we must first compute $\|b\|_2$. Taking the norm of both sides, we get $\|Z_i\|_2 = \left(\lambda_2 + \|b\|_2^{-1} \lambda_1\right) \|b\|_2$. Thus, $\|b\|_2 = \lambda_2^{-1}\left(\|Z_i\|_2 - \lambda_1\right)$. Plugging the last expression in (A3) and solving for $b$, we obtain $b^\star = \lambda_2^{-1}\left(1 - \|Z_i\|_2^{-1} \lambda_1\right) Z_i$, for $\|b^\star\|_2 \neq 0$. We need to take into account that $\text{dom}(p) = \text{range}(p^*)$ and vice-versa. In particular, $\|b^\star\|_2 \neq 0$ iff $\|Z_i\|_2 > \lambda_1$. Therefore, we have

$$b^\star = \begin{cases} \lambda_2^{-1}\left(1 - \|Z_i\|_2^{-1} \lambda_1\right) Z_i & \|Z_i\|_2 > \lambda_1 \\ 0 & o.w. \end{cases} . \tag{A4}$$

From (A2) we now need to compute $g(b^\star)$. If $\|Z_i\|_2 \leq \lambda_1$, then $g(b^\star) = 0$. If $\|Z_i\|_2 > \lambda_1$, after some algebraic manipulations, we obtain $g(b^\star) = (2\lambda_2)^{-1}(\|Z_i\| - \lambda_1)^2$. Finally, (A1) gives us the desired result

$$\pi^*(Z) = (2\lambda_2)^{-1} \sum_{i=1}^{p} \begin{cases} \left(\|Z_i\|_2 - \lambda_1\right)^2 & \|Z_i\|_2 > \lambda_1 \\ 0 & o.w. \end{cases} . \tag{A5}$$

### A.2. Proof of Proposition 2

Since $\pi(B) = \sum_{i=1}^{p} \pi(B_i)$ is a separable sum, from Beck (2017) Remark 6.7, we know

$$\text{prox}_{\sigma\pi}(B) = \left(\text{prox}_{\sigma\pi}(B_1), \ldots, \text{prox}_{\sigma\pi}(B_p)\right)^T . \tag{A6}$$

From Fan and Reimherr (2016), we have $\text{prox}_{\sigma\lambda_1\|\cdot\|_2}(B_i) = \left[1 - \|B_i\|_2^{-1} \sigma\lambda_1\right]_+ B_i$. From Beck (2017) 6.2.3, we have $\text{prox}_{(\sigma\lambda_2/2)\|\cdot\|_2^2}(B_i) = (1 + \sigma\lambda_2)^{-1} B_i$. Moreover, $\|\cdot\|_2^2$ is a proper closed and convex function, thus we can compose $\text{prox}_{(\sigma\lambda_2/2)\|\cdot\|_2^2}$ and $\text{prox}_{\sigma\lambda_1\|\cdot\|_2}$ as described in Parikh et al. (2014), obtaining the desired form

$$\text{prox}_{\sigma\pi}(B_i) = (1 + \sigma\lambda_2)^{-1} \left[1 - \|B_i\|_2^{-1} \sigma\lambda_1\right]_+ B_i . \tag{A7}$$

## A.3. Proof of Proposition 3

We first prove (b), i.e. $\bar{Z} = \text{prox}_{\pi^*/\sigma}\left(B/\sigma - X^T\bar{V}\right)$. If we compute the derivative of $\mathcal{L}_\sigma\left(Z \mid \bar{V}, B\right)$ with respect to $Z_i$ and we set it equal to 0, we obtain

$$B_i/\sigma - \left(X^T\right)\bar{V} - \bar{Z}_i = \nabla\pi^*(\bar{Z}_i)/\sigma \tag{A8}$$

We now use the sub-gradient proximal operators characterization (Correa et al., 1992):

$$u = \text{prox}_f(t) \ \text{ if and only if } \ t - u \in \partial f(u) . \tag{A9}$$

Considering $t = B_i/\sigma - \bar{V}^T X_{(i)}$, $u = \bar{Z}_i$, and $f = \pi^*/\sigma$, the right hand side of (A9) is true by (A8). The left hand side of (A9) gives us $\bar{Z}_i = \text{prox}_{\pi^*/\sigma}\left(B_i/\sigma - \bar{V}^T X_{(i)}\right)$. To conclude the first part of the proof just note that $\bar{Z} = (\bar{Z}_1, \ldots, \bar{Z}_p)^T$.

For the second part of the proof, we need to find $\psi(V) := \mathcal{L}_\sigma\left(V \mid \bar{Z}, B\right)$. First, note that by the Moreau decomposition (4) $\bar{Z} = B/\sigma - X^TV - (1/\sigma)\,\text{prox}_{\sigma\pi}\left(B - \sigma X^TV\right)$. Plugging this into (5), after some algebraic manipulations, we obtain

$$\psi(V) = h^*(V) + \pi^*(\bar{Z}) + \frac{1}{2\sigma}\sum_{i=1}^p \left\| \text{prox}_{\sigma\pi}\left(B_i - \sigma V^T X_{(i)}\right)\right\|_2^2 - \frac{1}{2\sigma}\sum_{i=1}^p \|B_i\|_2^2 . \tag{A10}$$

We now have to compute $\pi^*(\bar{Z})$. If we set $T = B - \sigma X^TV$, then $\pi^*(\bar{Z}) = \sum_{i=1}^p \pi^*\left(\text{prox}_{\pi^*/\sigma}(T_i/\sigma)\right)$. In particular

$$\text{prox}_{\pi^*/\sigma}(T_i/\sigma) = T_i/\sigma - (1/\sigma)\,\text{prox}_{\sigma\pi}(T_i) = \begin{cases} (1 + \sigma\lambda_2)^{-1}\left(\lambda_2 + \|T_i\|_2^{-1}\lambda_1\right)T_i & \|T_i\|_2 > \sigma\lambda_1 \\ T_i/\sigma & o.w. \end{cases} . \tag{A11}$$

Composing (A11) and (2), again after some algebraic manipulations, we get $\pi^*(\bar{Z}) = (\lambda_2/2)\sum_{i=1}^p \left\| \text{prox}_{\sigma\pi}\left(B_i - \sigma V^T X_{(i)}\right)\right\|_2^2$. Plugging this into (A10) concludes our proof.

## A.4. Proof of Theorem 1

Remember that $T = B - \sigma X^TV$ and $\hat{X} = X \otimes I_k$. To prove (i), we just take the gradient of $\psi(Y)$ with respect to $Y$, as given in (11). In particular, note that $\frac{\partial V^T X_{(i)}}{\partial V_j} = X_{ji}$, i.e. the element in the $j$-th row and $i$-th column of the matrix $X$, and therefore that

$$\nabla_V\left(\frac{1+\sigma\lambda_2}{2\sigma}\sum_{i=1}^p \left\| \text{prox}_{\sigma\pi}\left(B_i - \sigma V^T X_{(i)}\right)\right\|_2^2\right) = \begin{bmatrix} -\sum_{i=1}^p X_{1i}\,\text{prox}_{\sigma\pi}\left(B_i - \sigma V^T X_{(i)}\right) \\ \vdots \\ -\sum_{i=1}^p X_{mi}\,\text{prox}_{\sigma\pi}\left(B_i - \sigma V^T X_{(i)}\right) \end{bmatrix} = -X\,\text{prox}_{\sigma\pi}(T). \tag{A12}$$

Next, to prove (ii), note that $\hat{\partial}^2\psi(V)$ is the $nk \times nk$ symmetric matrix $\begin{bmatrix} \frac{\partial\psi}{\partial V_1\partial V_1} & \cdots & \frac{\partial\psi}{\partial V_1\partial V_n} \\ \vdots & \ddots & \vdots \\ \frac{\partial\psi}{\partial V_n\partial V_1} & \cdots & \frac{\partial\psi}{\partial V_n\partial V_n} \end{bmatrix}$. In particular, each block here is the $k \times k$ matrix

$$\frac{\partial\psi}{\partial V_t\partial V_s} = \begin{cases} I_k + \sigma\sum_{i=1}^p X_{ti}\partial\,\text{prox}_{\sigma\pi}(T_i)X_{si} & t = s \\ \sigma\sum_{i=1}^p X_{ti}\partial\,\text{prox}_{\sigma\pi}(T_i)X_{si} & t \neq s \end{cases} . \tag{A13}$$

Thus, we have

$$\hat{\partial}^2\psi(V) = I_{nk} + \sum_{i=1}^p \begin{bmatrix} X_{1i}\partial\,\text{prox}_{\sigma\pi}(T_i)X_{1i} & \cdots & X_{1i}\partial\,\text{prox}_{\sigma\pi}(T_i)X_{ni} \\ \vdots & \ddots & \vdots \\ X_{ni}\partial\,\text{prox}_{\sigma\pi}(T_i)X_{1i} & \cdots & X_{ni}\partial\,\text{prox}_{\sigma\pi}(T_i)X_{ni} \end{bmatrix} = I_{nk} + \hat{X}\partial\,\text{prox}_{\sigma\pi}(T)\hat{X}^T . \tag{A14}$$

We now need to show that $Q \in \partial\,\text{prox}_{\sigma\pi}(T)$. Note that $\partial\,\text{prox}_{\sigma\pi}(T)$ is a $pk \times pk$ block-diagonal matrix, since $\frac{\partial\,\text{prox}_{\sigma\pi}(T_i)}{\partial T_j} = 0$ for $i \neq j$. Let us focus on $T_1$, and let $t_1, \ldots, t_k$ be its $k$ elements. Then, $(\partial\,\text{prox}_{\sigma\pi}(T_1))_{ij} = \frac{\partial\,\text{prox}_{\sigma\pi}(t_i)}{\partial t_j}$, for

$i, j = 1, \ldots, k$. Specifically, for $\|T_i\|_2 \leq \sigma\lambda_1$, it is straightforward to see that $\partial \operatorname{prox}_{\sigma\pi}(T_1) = 0$. For $\|T_i\|_2 > \sigma\lambda_1$, knowing that $\frac{\partial \|T_1\|_2}{\partial t_i} = \|T_1\|_2^{-1} t_i$, after some algebraic manipulations we obtain

$$\frac{\partial \operatorname{prox}_{\sigma\pi}(t_i)}{\partial t_j} = (1 + \sigma\lambda_2)^{-1} \begin{cases} 1 - \sigma\lambda_1 \|T_1\|_2^{-1} + \|T_1\|_2^{-3} \sigma\lambda_1 t_i^2 & i = j \\ \|T_i\|_2^{-3} \sigma\lambda_1 t_i t_j & i \neq j \end{cases}. \tag{A15}$$

(A15) shows us that $P_1 = \partial \operatorname{prox}_{\sigma\pi}(T_1)$. Without loss of generality, we can do the same way for $T_2, \ldots, T_p$ and prove (iii). To conclude the proof of the theorem, we note that since $Q \in \partial \operatorname{prox}_{\sigma\pi}(T)$, then $I_{nk} + \sigma\hat{X}Q\hat{X}^T \in \hat{\partial}^2\psi(V)$, and from Hiriart-Urruty et al. (1984) we have $\partial^2\psi(V)\operatorname{vec}(D) = (I_{nk} + \sigma\hat{X}Q\hat{X}^T)\operatorname{vec}(D)$, for every $D$ in the domain of $V$

## B. Additional Simulation Results

We ran all simulations on a MacBookPro with 3.3 GHz DualCore Intel Core i7 processor and 16GB ram. We reran all `python` simulations using `openblas` and `mkl` as blas systems, with threads=1,2 and `openmp`, with threads=1,4. In all scenarios, times match those reported in the paper that are obtained considering `openblas` with 2 threads and `openmp` with 4 threads. The following versions of `sklearn` and `glmnet` are used: `scikit-learn==0.22.2` and `glmnet==4.1`

*Table B.1.* $a$, $b$ and $c$ report *mean* CPU time in seconds for `fgen`, `sklearn` and `glmnet`, respectively, over 20 replications of the same simulation scenari. In parenthesis we report *standard errors*. For each scenario we consider three values of $c_\lambda$, which are held fixed over the replications.

| $\alpha = 0.8$, $l = 0.25$ | | | n=1000 | | | n=5000 | | |
|---|---|---|---|---|---|---|---|---|
| $p; p_0$ | k | $c_\lambda$ | a | b | c | a | b | c |
| | | 0.8 | **0.2** (0.02) | 1 (0.03) | 0.7 (0.01) | **0.6** (0.01) | 9.1 (0.24) | 3.1 (0.03) |
| $2(10^4); 10$ | 5 | 0.4 | **0.3** (0.00) | 1.0 (0.01) | 0.6 (0.00) | **1.4** (0.04) | 9.5 (0.28) | 3.2 (0.04) |
| | | 0.2 | **0.4** (0.01) | 1.2 (0.04) | 0.6 (0.02) | **1.3** (0.04) | 10.2 (0.13) | 3.0 (0.10) |

*Table B.2.* $a$, $b$ and $c$ report CPU time in seconds for `fgen`, `sklearn` and `glmnet`, respectively. The full $c_\lambda$ grid consists of 100 log-spaced points between 1 and 0.01. We truncate the path search when $max$ active components are selected. $runs$ is the corresponding number of explored $c_\lambda$ values. We fix 1000 seconds as time limit.

| $\alpha = 0.8$, $l = 0.25$ | | | n=500 | | | | n=1000 | | | | n=5000 | | | |
|---|---|---|---|---|---|---|---|---|---|---|---|---|---|---|
| $p; p_0$ | k | max | runs | a | b | c | runs | a | b | c | runs | a | b | c |
| | | 5 | 7 | **1.6** | 26.2 | 3.2 | 8 | **3.8** | 63.3 | 7.4 | 6 | **11.0** | 399.5 | 30.5 |
| $10^5; 10^2$ | 5 | 20 | 12 | **2.8** | 45.4 | 5.2 | 14 | **5.2** | 111.1 | 10.9 | 15 | **26.7** | >1000 | 59.1 |
| | | 100 | 16 | **5.1** | 64.7 | 6.3 | 24 | **11.7** | 196.1 | 18.0 | 41 | **90.1** | >1000 | 146.7 |