# OpenReview forum: "A Highly-Efficient Group Elastic Net Algorithm with an Application to Function-On-Scalar Regression"
_NeurIPS.cc/2021/Conference — NeurIPS 2021 Poster_

### Official Review · Reviewer_nVk8 · 2021-07-09

**Rating:** 4
**Confidence:** 3

**Summary:**

The paper proposes a new group elastic net algorithm solver for linear regression problem with multivariate response variable. The paper then claims its application on function-on-scalar.

**Limitations And Societal Impact:**

The authors claim that they do not see any potential for adverse societal impacts. Since this paper is mainly theoretical, I agree with the authors' claim.

**Main Review:**

I am not an expert in optimization, but I feel that this paper is strikingly similar to [1], which aims to solve a similar elastic net problem but with univariate response variable. While the paper extends the univariate case to multivariate case, I think this extension is not very complicated. Most of methods and results have similar corresponding in [1]. At least for me, this paper seems trivial. I think that the authors should give a thorough related work section and explain what their main contributions are.

Besides, I feel that the application on function-on-scalar regression is also a trivial application. The contribution of using FPCA is overly stated, since it seems very natural and is not the only choice. In fact, by using any function basis and doing dimension reduction via projection, we can always easily convert a function-on-scalar problem to a linear regression problem with multivariate response.

The paper is technically sound. The results are well proved.

The paper is also clearly written and well-organized.

In terms of significance, I think the paper has its own value in application perspective. Since the problem solved by the paper can appear in many places, an efficient implementation will be valuable. However, my main concern is about its limited originality as mentioned above. I would suggest this paper to be submitted to some statistics software focused conferences/journals rather than methodology/theory focused ones like NeurIPS. If the authors insist, it is better to explain its novelty in terms of methods or theoretical analysis.


**Time Spent Reviewing:**

4

---

> ### Author Response · Authors · 2021-08-10
> **First Reply**
>
>
> "I am not an expert in optimization, but I feel that this paper is strikingly similar to [1], which aims to solve a similar elastic net problem but with univariate response variable. While the paper extends..."
>
> This comment does not specify which reference is [1], however we would like to politely disagree and stress that extending the algorithm from the univariate to the multivariate case is not at all trivial since the optimization must be done jointly across coordinates of the outcome, and can't be separated into separate subproblems for each coordinate.  It involves defining the proximal operator, the conjugate function of $\pi$, and its gradient and Hessian in a more general framework -- working with matrices and higher order tensors as opposed to simple vectors. This requires, for example, usage of Kronecker products and block diagonal matrices, and rewriting tensors in matrix form to preserve the sparsity of the problem, without blowing up the dimension of the algorithm. The definitions of all these operators and of their derivatives, as far as we know, are new in the literature. We also point out that, in addition to the current proposal, they may be used in other optimization algorithms based on augmented Lagrangians.
>
> .
>
> "Besides, I feel that the application on function-on-scalar regression is also a trivial application. The contribution of using FPCA is overly stated, since it seems very natural...
>
> However, my main concern is about its limited originality as mentioned above. I would suggest this paper to be submitted to some statistics software focused conferences/journals rather than methodology/theory focused ones like NeurIPS. If the authors insist, it is better to explain its novelty in terms of methods or theoretical analysis"
>
> We agree with the Reviewer that other bases can be readily used in fgen for function-on-scalar regressions. In order to preserve the algorithm's efficiency, the  basis system needs to be orthonormal (otherwise (18) would not hold and computing integrals would have a much higher computational cost). Among the many orthonormal bases one could envision, FPCA has the advantage of being the most parsimonious -- in the sense that, by construction, it allows one to reconstruct the response curves using fewer coefficients than any other orthonormal basis, thereby reducing the dimension of each group $k$ and thus the computational burden.
>
> We also believe that, whether leveraging FPCA or potentially other orthonormal bases, the extension we propose is valuable from a methodological point of view exactly because of its broad application scope. Many contemporary studies can benefit from effective and efficient feature selection for functional linear models. For example, Reviewer 2b9Y suggests applications to the construction of polygenic risk scores and the analysis of gene expression data.

---

> > ### Comment · Reviewer_nVk8 · 2021-08-29
> > **Thanks for the response**
> >
> > My apologies that I forgot to include the reference. By [1] I am referring to the paper "An Efficient Semi-smooth Newton Augmented Lagrangian Method for Elastic Net", by Tobia Boschi, Matthew Reimherr, Francesca Chiaromonte.
> >
> > While the authors kindly point out their contributions, I still have the impression that these extensions to multivariate case may not be challenging enough given the above work. I would like to ask if the authors can demonstrate what are the main challenging parts of your work compared with [1]? I am open to raise the score based on your answer. Thanks.

---

> > > ### Author Response · Authors · 2021-08-30
> > > **Second Reply**
> > >
> > > Thanks for clarifying reference [1]. We will try to explain clearlier the main challenge with respect to [1].
> > >
> > > Including a group structure in the response, increases the dimension of the optimization problem. Indeed, while the least square part of the minimization problem and the $l2$ penalty part can be separated (since they involve squared norms), the presence of the $l1$ penalty requires one to carry out a joint minimization. One has to redefine a set of all-new operators starting from a global penalty, which is the sum of the $l1$ and $l2$ penalties.
> > > In particular, the definition of proximal operator of $\pi$ and the conjugate function $\pi^*$, the computation of the gradient, and the Hessian of the Lagrangian function are the main mathematical challenges. One has to carefully work out the necessary tensors and block matrices in order to preserve the computational efficiency of the problem.
> > >
> > > For example, a key aspect of our derivation is that though we start with a non-separable objective function, our derivation of $\pi^*$ does separate, allowing us to rewrite the Hessian operator, which is a third-order tensor, as a diagonal block matrix, which exploits the sparsity of the problem for rather dramatic computational gains.

---

> > > > ### Comment · Reviewer_nVk8 · 2021-09-02
> > > > **Thanks for reply**
> > > >
> > > > Thanks for your reply. Now I can better understand what are the challenges of your extension. Based on your contributions, I think the paper deserves a higher score; however, I still have my concern about whether this contribution is novel enough to be published on a top conference like neurips. I will change my score accordingly.

---

### Official Review · Reviewer_FbcU · 2021-07-15

**Rating:** 6
**Confidence:** 3

**Summary:**

The main contribution of the paper is two-fold.
- First, the paper proposed a fast algorithm to learn the regression coefficients with the group elastic net regularization. Suppose that the dataset has k-dimensional response variables, and therefore the regression coefficients B has the form $B\in R^{p\times k}$ (p: number of features). Then the group elastic net regularization enhances the sparsity of features for all k response variables (i.e., some rows in B are to be zero).
- Secondly, the paper proved that it can be extended to function-on-scalar setup, that is, we assume a regression problem where the response variable is a functional object of summing $k$ basis functions. (Therefore each element in $B\in R^{p\times k}$ is a functional object.)


**Limitations And Societal Impact:**

As stated in the main review, the proposed method can largely reduce the learning cost both theoretically and experimentally. So it seems to be of use.

The limitation of the method is discussed in Section 5. As possible extensions of the method, the authors consider (1) different number of coefficients in a group, and (2) use for other functional models than function-on-scalar model.


**Main Review:**

The proposed method is to efficiently learn a sparse function-on-scalar model, and (although it is not specifically stated in the paper) a sparse multitask model. It is theoretically proved that, even if the number of features is huge, the main computational cost depends on the number of only a part of features ($|\mathcal{J}|$ in the paper), which is of value.

Suggestions:

-   The improvements from the conventional method is better to be stated in Section 1. For example, the paper stated three conventional methods in line 152 and their limitation for our problem setup, but the overview of such discussions is better to be appeared in Section 1 to emphasize how the proposed method achieved the improvements.
-   The paper mainly consider a fast method to learn a sparse function-on-scalar model, but it solves a sparse multitask model (multiple response variables) as a building block. So, in order to clarify the advantage of the proposed method, it is better to discuss the relationships to conventional learning methods for sparse multitask model. For example, the following paper discusses the group lasso (instead of group elastic net) regularization for multitask learning:
    -   Karim Lounici, Massimiliano Pontil, Alexandre B. Tsybakov and Sara A. van de Geer, "Taking Advantage of Sparsity in Multi-Task Learning". COLT2009.

Questions:

-   Suppose that we consider a model with ordinary response variable (i.e., non-multitask, non-function-on-scalar). Then, does the proposed method work efficiently compared to conventional elastic net learning methods? In other words, how the multitask setup and/or function-on-scalar setup is employed for efficient computation?
-   In Section 3.1, the definition of $\hat{X}_\mathcal{J}$ is not so obvious: since $T\in R^{p\times k}$, $\mathcal{J}$ must be $\mathcal{J}\subseteq\{1, 2, \dots, p\}$ and therefore inappropriate as a subscript for $\hat{X}\in R^{nk\times pk}$. Does it mean that $\hat{X}_\mathcal{J} = \hat{\mathcal{X}}$ where $\mathcal{X} = X_\mathcal{J}$?


**Time Spent Reviewing:**

3

---

> ### Author Response · Authors · 2021-08-10
> **Frist Reply**
>
> "The improvements from the conventional method is better to be stated in Section 1. For example, the paper stated three conventional methods in line 152 and their limitation..."
>
> We will add a paragraph to Section~1 briefly describing how our approach differs from the existing methods in the literature.
>
> .
>
> "The paper mainly consider a fast method to learn a sparse function-on-scalar model, but it solves a sparse multitask model (multiple response variables) as a building block. So, in order to..."
>
> We thank the Reviewer for this important suggestion. We will add a paragraph text comparing our method to sparse multitask models and our method to in Section~1. We will also add some key multitask models references, such as:
> - Zhang, Yu, and Qiang Yang. "An overview of multi-task learning." National Science Review 5.1 (2018): 30-43.
> - Zhou, Jiayu, Jianhui Chen, and Jieping Ye. "Malsar: Multi-task learning via structural regularization." Arizona State University 21 (2011).
>
> .
>
> "Suppose that we consider a model with ordinary response variable (i.e., non-multitask, non-function-on-scalar). Then, does the proposed method work efficiently compared to conventional elastic net learning methods? In other words, how the multitask setup and/or function-on-scalar setup is employed for efficient computation?"
>
> If we consider an ordinary response variable, we can still run fgen setting $k = 1$, i.e.~restricting each group to a single element. In this case, the method would solve the classical Elastic Net problem, but still using the Augmented Lagrangian and Semi-Smooth newton methods and exploiting their sparsity. Also this restricted implementation has a strong computational advantage with respect to its competitors, as shown in Li, X., D. Sun, and K.-C. Toh (2018) (Lasso case) and Boschi, T., M.
> Reimherr, and Chiaromonte, F. (2020) (Elastic Net case).
>
> .
>
>
> "In Section 3.1, the definition of  $\hat X_\mathcal{J}$ is not so obvious..."
>
> We thank the Reviewer for pointing this out. We will correct the definition of $\hat X_\mathcal{J}$ as follows:
>
> - $\hat X_\mathcal{J} = X_\mathcal{J} \otimes I_k$,
>
> where $X_\mathcal{J} \in \mathbb{R}^{n \times r}$ is the sub-matrix of $X$ restricted to the columns in $\mathcal{J}$. So, we have $\hat X_\mathcal{J} \in \mathbb{R}^{nk \times rk}$.

---

### Official Review · Reviewer_2b9Y · 2021-07-19

**Rating:** 6
**Confidence:** 2

**Summary:**

This paper adds to the various existing solvers for group elastic net regression. Here, a grouping of input variables is jointly penalized with both L1 and L2 penalties in order to yield an estimate of sparse penalized regression coefficients. The newly developed optimization algorithm — semi-smooth newton augmented langrangian — incorporates group structure and leverages sparsity to solve the optimization problem an order of magnitude faster than off-the-shelf methods from python sklearn and R glmnet. The method is further extended to function-on-scalar regression by leveraging functional principal components analysis. This extension allows for analyzing real data where the target is functional curves, in this case growth curves of childhood obesity development. In the INSIGHT study, one single nucleotide polymorphism was found to be significantly associated with childhood obesity, a locus occurring in a gene that has been previously associated with anthropometric-related phenotypes.

**Limitations And Societal Impact:**

The authors have addressed limitations of their method. Societal impacts of the GWAS application should be discussed a bit more thoroughly considering this methodology paves the way for more function-on-scalar type GWAS to be investigated, and the current assumptions of the method should be made clear for those interested in hypothesis testing.

**Main Review:**

The authors present a novel algorithm for fast optimization of group elastic net regression. This is further extended to function-on-scalar regression to increase the relevance of the method. The motivation is clear and the paper is well-organized and clearly written. The experiments are clearly laid out and easy to follow. The empirical results comparing computational efficiency are strong, and it seems the new methodology here could become a new baseline, given its comparison to already highly adopted packages. Theoretical results further confirm the efficiency of the proposed method. The applications to a childhood obesity GWAS provide a novel perspective on traditional GWAS framework, and while the result of confirming a previously discovered significant association is encouraging, there are some areas that could be improved to increase impact.

First, GWAS is not typically considered with penalized regression due to identifiability issues. For this reason, it may be more applicable to consider this methodology for prediction rather than hypothesis testing, which is difficult to interpret in this setting. The signal-to-noise ratio in GWAS is often very low (as defined by the overall variance explained or variance explained per SNP), and it appears the current methodology might struggle in this setting as well. In particular, there is only a single SNP that’s identified by the proposed method, and the $c_\lambda$ parameter is effectively at its maximum, so the model is being constrained as heavily as possible. It might be more fruitful to consider applications such as polygenic risk scores (where prediction is considered), or to look into gene expression experiments with measurements taken over time in order to increase the sample size. The sample size here (n=210) is bound to be underpowered to detect associations.


**Time Spent Reviewing:**

4

---

> ### Author Response · Authors · 2021-08-10
> **First Reply**
>
> "First, GWAS is not typically considered with penalized regression due to identifiability issues. For this reason, it may be more applicable to consider this methodology for prediction rather than hypothesis testing, which is difficult to interpret in this setting. The signal-to-noise ratio in GWAS... "
>
> We agree with the Reviewer; due to high-feature collinearity, low signal-to-noise ratio, and ultra-high dimensionality GWAS data are very hard to analyze. While being well aware of all its complexities and potential pitfalls, our main aim in presenting a GWAS analysis is to advertise the computational efficiency and the broad applicability of fgen. We do not aim to to make any strong domain conclusions, but since the selected SNP appears to be relevant in the literature, we point that out. We will further stress the complexities of GWAS in the revision.
> We also thank the Reviewer for the suggestion of considering polygenic risk scores and gene expression experiments. We will investigate these applications in future work.
>
> .
>
> "The authors have addressed limitations of their method. Societal impacts of the GWAS application should be discussed a bit more thoroughly considering this methodology paves the way..."
>
> We will add some text in the revision to stress the complexities of GWAS (see also above). In particular, we will add text to the Conclusions section, discussing the difficulty of interpreting GWAS and their potential societal impacts. We will also state more clearly key assumptions in the Introduction. In particular, we will state how feature selection methods (including our proposal) rely critically on some bounds on the amount of feature collinearity. This may in fact be problematic in GWAS, so users should be very careful in interpreting
> results; e.g., selected SNP's may just be proxies for other, causal SNP's in their vicinity.

---

### Official Review · Reviewer_ZQkA · 2021-08-02

**Rating:** 7
**Confidence:** 4

**Summary:**

This paper proposed a novel optimization algorithm for the Group Elastic Net problem, with additional application to the function-on-scalar regression problem. By exploiting the sparsity structure of the Augmented Lagrangian, the new algorithm significantly reduces the computational load. Experiments also showed supreme results, comparing with multi-task elastic net procedures in sklearn and glmnet. The paper is well written and easy to follow.

**Limitations And Societal Impact:**

Here are a few minor concerns I have for the paper:

1. Table I only showed the speedup compared with sklearn and glmnet, but no comparison in solution accuracy is provided along with it. The speedup is meaningful only if the solution accuracy is comparable to other methods. A better way to compare different algorithms is to show the accuracy (or error) along with the time spent in the computation.

2. The discussion on the convergence rate could be more rigorous. For example, the super-linear convergence rate is for the semi-smooth Newton subproblem, and I did not see any strict theorem or proof of convergence rate for the whole algorithm.

3. The augmented Lagrange algorithm converges as the \sigma variable goes to infinity. How is this increase in \sigma scheduled in the algorithm? Will different scheduling scheme impact the convergence property?

**Main Review:**

The following contributions from the paper are quite novel:
* Exploiting the analytical form of the Fenchel conjugate of the group elastic net regularization.
* Using semi-smooth Newton method to solve the dual variable in the subproblem.
* Applying the algorithm to function-on-scalar regression problem.

This novel optimization algorithm proposed in this paper can be very valuable to the machine learning practitioners.

**Time Spent Reviewing:**

4 hours

---

> ### Author Response · Authors · 2021-08-10
> **First Reply**
>
> "Table I only showed the speedup compared with sklearn and glmnet, but no comparison in solution accuracy is provided along with it. The speedup..."
>
> fgen, glmnet, and sklearn solve the same convex minimization problem and therefore converge to the same solution. This is true in principle, but we also verified it numerically in all of the experiments presented in the paper (we also made sure all used the same convergence tolerance). For this reason, we did not present comparisons of prediction or accuracy performance.
> We will clarify this point in the revision.
>
> .
>
> "The discussion on the convergence rate could be more rigorous. For example..."
>
> We agree we should have been more precise when discussing the convergence rate for the whole algorithm. Following the existing literature -- Tomioka, R. and M. Sugiyama (2009) and Li, X., D. Sun, and K.-C. Toh (2018) -- the convergence rate of the entire algorithm is the sum of the convergence rate of the two subproblems, which is still super-linear. We will add a sentence clarifying this.
>
> .
>
> "The augmented Lagrange algorithm converges as the $\sigma$ variable goes to infinity. How is this increase in $\sigma$ scheduled in the algorithm? Will different scheduling scheme impact the convergence property?"
>
> We describe our choice of $\sigma$ on line 253. We start from an initial value of $\sigma^0 = p_0/p$ and increase it by a factor of 5 every iteration. A similar choice of $\sigma$ is made by Tomioka, R. and M. Sugiyama (2009). If $\sigma^0$ is too large, or we increase it too quickly, fgen does not converge to the optimal solution. Vice-versa, if we increase $\sigma$ too slowly, the algorithm needs more iterations to converge.

---

### Decision · Program_Chairs · 2021-09-28

**Decision:**

Accept (Poster)

**Comment:**

Three reviewers indicated acceptance, only one reviewer had concerns about the novelty with respect to reference [1]. It turned out, however, that [1] is basically an unpublished pre-print of this paper, and in their rebuttal, the authors could convincingly show that even with respect to [1], the paper contains some important extensions. So I recommend acceptance of this paper.

**Consistency Experiment:**

NeurIPS has a long history of experimentation. In 2014, NeurIPS ran an experiment in which 10% of submissions were reviewed by two independent committees to quantify the randomness in the review process. This year, we repeated a variant of this experiment to see how the quality of the review process has changed over time.  This paper was part of the experiment and was therefore assigned to two committees (consisting of reviewers, an Area Chair, and a Senior Area Chair) that reached independent decisions.  If both committees made the same recommendation, this recommendation was followed. If a single committee recommended acceptance, the paper was accepted (with the exception of a few cases in which the other committee identified what we considered a fatal flaw, e.g., an error in a key result).

Both committees reached the same decision: **Accept (Poster)**

The other committee assigned to the paper recommended **Accept (Poster)**.  You can find the other set of reviews, along with any follow up discussion with the authors here:
https://openreview.net/forum?id=U7M1yp1vJ8